# Intake of Radionuclides in the Trees of Fukushima Forests 3. Removal of Radiocesium from Stem Wood, *Cryptomeria Japonica* (L.f.) D. Don. †

**Tomoko Seyama [1], Ryohei Arakawa [2], Shogo Machida [2], Sota Yoshida [2], Akihiko Maru [3], Kei'ichi Baba [4], Yoshinori Kobayashi [5], Rumi Kaida [2], Teruaki Taji [2], Yoichi Sakata [2], Tomoaki Iijima [1] and Takahisa Hayashi [2,*]**

[1]  Department of Forest Science, Tokyo University of Agriculture, Tokyo 156-8502, Japan; t3seyama@nodai.ac.jp (T.S.); tomo-iijima@jcom.home.ne (T.I.)
[2]  Department of Bioscience, Tokyo University of Agriculture, Tokyo 156-8502, Japan; goyen0514@gmail.com (R.A.); grow.to-ward-ly.key@ezweb.ne.jp (S.M.); boruto8401sota@ezweb.ne.jp (S.Y.); r3kaida@nodai.ac.jp (R.K.); t3teruak@nodai.ac.jp (T.T.); sakata@nodai.ac.jp (Y.S.)
[3]  Zoetis Japan Plant Health Division, Tokyo 151-0053, Japan; maru3@gmail.com
[4]  Research Institute for Sustainable Humanosphere, Kyoto University, Uji 611-0011, Japan; kbaba@rish.kyoto-u.ac.jp
[5]  K's Wood Laboratory, Nara 634-0804, Japan; y.koba@coda.ocn.ne.jp
*   Correspondence: takaxg@nifty.com
†   This paper is dedicated to Peter Albersheim who died on 23 July 2017.

**Abstract:** Nuclear power plant accidents have dispersed radiocesium into the atmosphere to contaminate trees with no turnover in heartwood, as occurred in Fukushima, and as has persisted for over 30 years around Chernobyl. Here we employ the ponding method, in which radiocesium can be flushed out from the cross-cut edges of Japanese cedar, *Cryptomeria japonica* (L.f.) D. Don., stem with water due to xyloglucan degradation in tracheids. Furthermore, lab-scale ponding experiments have shown that a non-detectable level of radiocesium has been observed not only in the pool water used for 575 days but also in the water containing recombinant xyloglucanase. This traditional technology is now a new biotechnology.

**Keywords:** Japanese cedar; radiocesium removal; ponding method; tracheid; pit membrane; xyloglucan

---

## 1. Introduction

The meltdown and subsequent hydrogen explosion at the Fukushima nuclear power plant in Japan in March 2011 dispersed tremendous amounts of radionuclides into humans, forests, and other objects [1]. The fallout was primarily in the form of radiocesium, which is the most damaging to forest trees; cesium-134 and cesium-137 have been estimated at $1.8 \times 10^{16}$ and $1.5 \times 10^{16}$ Bq, respectively [2,3]. These radiocesiums were dispersed by the northwest wind-flow and deposited on trees in significant concentrations over approximately 200,000 hectares of rural land, more than 90% of which is forested. The dispersed radiocesium landed in aerosol form on the outer surface of the bark and migrated directly into entire tree bodies through translocation to the sapwood and heartwood [4,5]. Although radiocesium is recycled among other forest components—such as living plant tissues, leaf litter, mulch, and soil—once it enters the heartwood, there is no more turnover, and it continues to accumulate [6]. This has made it very difficult to remove radiocesium from living tree plants, though we attempted to use a foliar spray of potassium solution applied by hand and by radio-controlled helicopter in a small forest, as well as the direct injection of potassium solution into tree trunks over several years

in order to reduce radiocesium through the use of potassium fertilization [7], see Figure S1. None of these approaches, however, succeeded in decreasing the radiocesium levels in Fukushima forest trees.

To date, no permissible level has been set for wood products in Japan, although it is now illegal to sell any timbers in Fukushima with more than 1000 cpm on its surface, as measured by a NaI(TI) scintillation counter [8]. Unfortunately, such blanket regulatory rules do not accurately reflect the amount of radiocesium in wood: 100 cpm in the heartwood is equivalent to 1055 Bq/kg and 650 cpm in the bark to 33,400 Bq/kg, as shown in Figure S2. However, 1000 Bq/kg in wood has been set as the legal limit in Ukraine and other countries affected by the Chernobyl disaster. Even today, more than 30 years after the Chernobyl disaster, a biomass energy company in Sweden continues to monitor radiocesium levels in the wood and bark from various Swedish forests because it is difficult for the systems in their plants to capture radiocesium fly ash from biomass sources with levels higher than 1000 Bq/kg [9]. Since cesium-137 has a long half-life ($30.17 \text{ yr}^{-1}$), the radiocesium will remain in the trees throughout their entire lives.

There is no method to treat contaminated wood. Here we employ the ponding method of wood impregnation to remove radiocesium from tree stems through water flow [10–12]. This procedure has traditionally been used to increase the water-permeability of water-rich stem wood. Initially, freshly cut stems are placed in a pond, river, or sea in order to introduce bacteria that can degrade the pit membranes in tracheids or vessels over the course of several years. The stems are then positioned vertically to allow the water to drain out, a process through which the bacteria can also flow out. This is also an effective method for drying stem wood without warping or cracking it [13]. It should be noted that the pit membrane as a target is composed of a primary wall, mainly xyloglucan and cellulose.

## 2. Materials and Methods

### 2.1. Preparation of Stems for Ponding Method

All tree stems used were obtained from private plantation forests in Fukushima, where standing Japanese cedar trees in Minamisoma forest (37°31′45.6″N 140°56′22.3″E) were cut into trunks 2 m, 3 m, and 4 m long.

The trunks were immediately ponded along with 20 kg of zeolite in a pool of water obtained at 37°33′27.0″N 140°55′35.8″E, as shown in Figure S3 and Table S1. The diameters of the stems were approximately 41 to 50 cm at breast height. Six of the 2-m stems (~1200 Bq/kg in the xylem) were removed from the pool after 165 days of ponding and six more stems of the same properties were removed after 575 days. Stems of each subsample were allowed to drain for 1 month whereby three stems were positioned vertically, while the other three were laid out horizontally. After 165 days of ponding, six of the 2-m stems (~1200 Bq/kg in the xylem) were removed from the pool and allowed to drain for 1 month, three of which were positioned vertically while the other three were laid out horizontally. Both cross-cut sections (20 cm width) were used as non-treated controls.

The heartwood sections (1960 Bq/kg) were sterilely excised in small pieces (16 cm$^2$ × 10 cm in length) from the Japanese cedar stems with 50 g of zeolite and subjected to the ponding method at 25 °C in various incubation mixtures (1 L). The waters of the pool used for 575 days described above and the water of the river in Fukushima was obtained at 37°33′27.0″ N 140°55′35.8″ E, respectively. Recombinant xyloglucanase was expressed in *Escherichia coli* cells harboring the pET-32 Xa/LIC expression vector fused with the full-length cDNA for *AaXEG2* (accession number AF043595). The xyloglucanase solution (1000 units in 1 mL of 0.1 M MES/KOH, pH 6.2) was added twice to the pieces in 1 L of distilled water. The enzyme preparation specifically hydrolyzed the 1,4-β-glucan of xyloglucan but not those of carboxymethyl-cellulose and cello-oligosaccharides. After 84 days of immersion, the sections were removed from the mixture and allowed to drain vertically for 2 weeks.

## 2.2. Determination of Radioactivity

Drained stem trunks were each cut transversely into 20-cm-thick trunks after removal of the bark and subjected to autoradiography with an Image Analyzer (Fujifilm, Tokyo, Japan) to confirm the effect of decontamination inside the trunks. Image plates were attached to the sample sections for 3 weeks. No signals were observed from the stored trunks that were cut before the accident.

The 20-cm-thick trunks were divided into their sapwood and heartwood, which were smashed to determine their radioactivity in a glass vial. The radiation in each fraction was determined using either an AlokaAccuFles γ7001 scintillation counter (Aloka, Tokyo, Japan) or, for low radioactivity, Canberra germanium detectors (Canberra, Meriden, USA).

## 2.3. Microscopy

The 5-mm sections (0.5 cm × 0.5 cm × 0.5 cm) of stem trunks were prepared for microscopic observation to analyze the structural changes. Sections were fixed in 3% glutaraldehyde in 70 mM sodium phosphate buffer (pH 7.0) overnight at 4 °C and then sliced into 200 μm-thick longitudinal sections using a freezing/sliding microtome at −20 °C. The sections were incubated with 50% sodium hypochlorite for 10 min to remove the protoplasm. After five 10-min washes with sodium phosphate-buffered saline (PBS), the sections were incubated in a blocking solution of PBS containing 1% bovine serum albumin for 1 h. They were then incubated in a 20-fold dilution of anti-xyloglucan antibody CCRC-M1 (CarboSource, Athens, USA) for 2 h and then washed with PBS containing 0.5% Tween-20 [14]. The sections were further incubated in a 50-fold dilution of anti-mouse IgG antibody conjugated with Alexa 488 (Molecular Probes, Eugene, USA) for 2 h and embedded in an antifading reagent (Molecular Probes). They were then observed under a microscope equipped with a confocal laser-scanning system (LSM 780; Carl Zeiss, Munchen, Germany). Alexa 488 fluorescence was observed with excitation at 488 nm (Ar laser) and emission at 495 to 540. The instrument settings were unchanged when comparing fluorescence levels between samples.

## 2.4. Determination of Xyloglucanase Activity

One 2-m stem ponded for 165 days was cut into 20-cm-thick stems to analyze the relationship between xyloglucanase activity and decontamination. The inside of the xylem for each trunk was homogenized with liquid nitrogen, and the resulting powder was ground with 20 mM sodium phosphate buffer (pH 6.2) containing 1 M NaCl in a mortar. The extract supernatant was adjusted to 80% saturation with solid ammonium sulfate to generate a precipitate which was solubilized in 20 mM phosphate buffer (pH 6.2). Xyloglucanase activity was assayed viscometrically using amyloid xyloglucan as the substrate and Cannon-Manning semi-micro viscometers [15]. One viscosity unit of activity is defined as the amount of enzyme that causes, during the early stages of the reaction, a 1% decrease in the relative viscosity of 0.9 mL mixtures of 0.5% (w/v) amyloid xyloglucan in 2 h at 35 °C.

## 2.5. Potassium Fertilization

Potassium (K) treatments were included in all trials to identify their antagonistic effectiveness in suppressing radiocesium uptake [7]. Concentrations of between 0.5 and 2M of potassium chloride were used. The direct injection of potassium into tree trunks was accomplished by making a 0.6 mm hole at a depth of 5 to 9 cm in the stems and inserting the nozzle of a bottle containing potassium chloride solution.

## 3. Results and Discussion

### 3.1. Ponding Experiments in Fukushima Water Pool

In the vertical stems, radiocesium was less abundant in the bottom portion of the stem than in the top portion, although the level of radiocesium in the top portion was also decreased, as shown

in Figure 1. In the horizontal stems, radiocesium levels at the cross-cut edges of the stem were lower than before treatment but remained higher than those in the vertical stems. It appears that radiocesium was gradually removed from both edges of the stems through water penetration. This process caused a decrease in radiocesium in both the sapwood and the heartwood. After 575 days of ponding followed by vertical draining, radiocesium was almost completely removed from the 2-m stems, the level of which had dropped to approximately 41 Bq/kg in the wood. The level would have been decreased even further had the use of a circulating water system been possible. When longer stems (4 m) were left in the pool for 575 days, a certain level of radiocesium remained in the middle portions of such stems, see Figure 2, similar to the results shown in the 2-m stems for short ponding (165 days). In ponding, the longer the stem length, the more days required to reduce radiocesium levels. Nevertheless, the pattern of radiocesium remaining in each stem was almost similar to the relative distributions between low (~1200 Bq/kg), medium (~2500 Bq/kg), and high (~4100 Bq/kg) levels in the xylem. The level of decontamination was not dependent on the content of radiocesium but instead on the length of the stem and the number of ponding days.

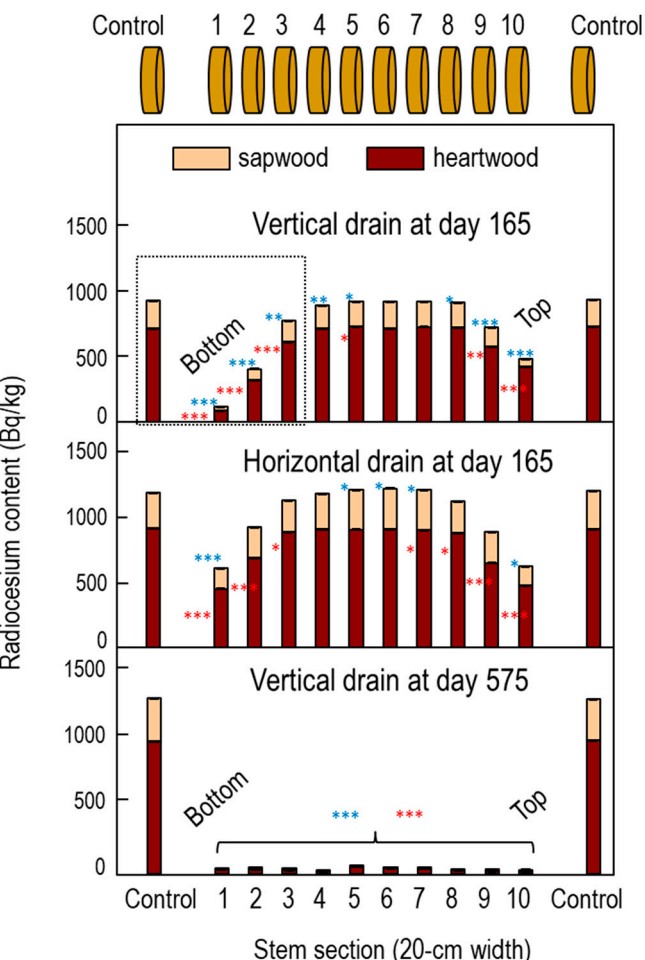

**Figure 1.** Ponding experiments for the stems of Japanese cedar from Fukushima forest. Level of radiocesium decontamination in 2-m stems. The dotted box was further analyzed in Figure 3. Bar plot the average ±SD of the data from three measurements. Significance at the * $p < 0.05$, ** $p < 0.01$, and *** $p < 0.001$ levels compared with non-treated stems (red, heartwood; blue, sapwood) in Student's t-test and indicated.

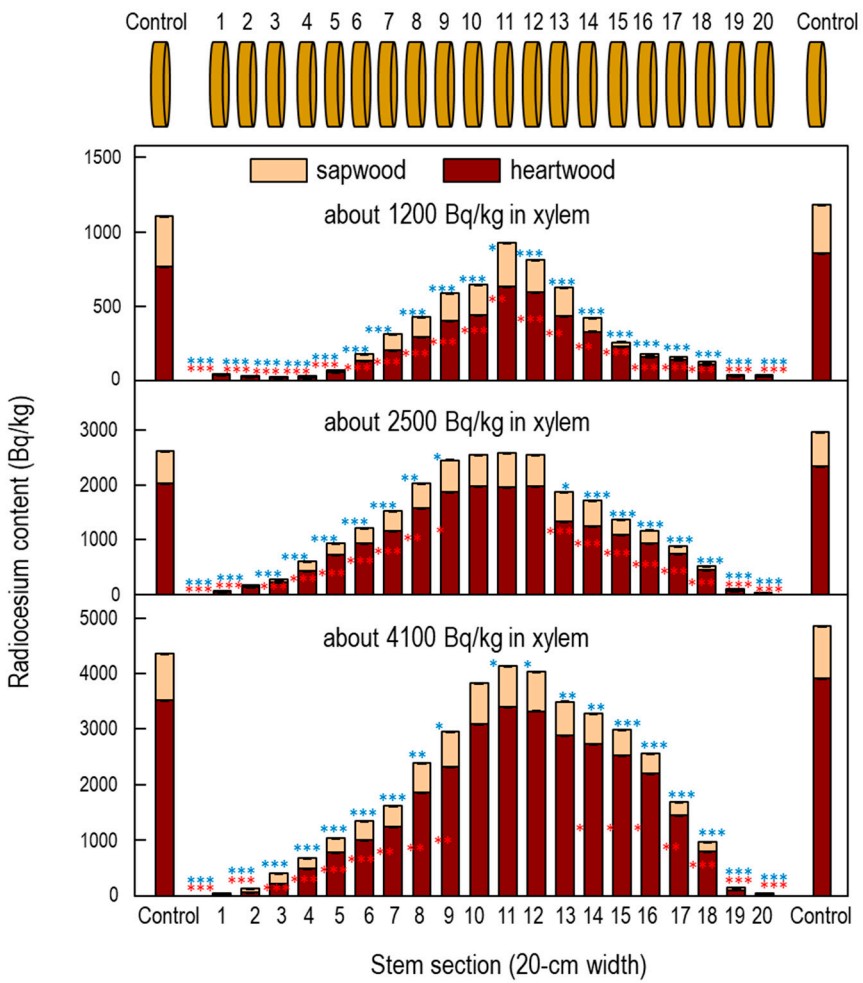

**Figure 2.** Ponding experiments for the stems of Japanese cedar from Fukushima forest. Bar plot the average ±SD of the data from three measurements. Significance at the * $p < 0.05$, ** $p < 0.01$, and *** $p < 0.001$ levels compared with non-treated stems (red, heartwood; blue, sapwood) in Student's t-test and indicated.

### 3.2. Analysis at the Cross-Cut Edge of Stem

The immuno-fluorescence microscopy of tracheids, as shown in Figure 3, shows high levels of xyloglucan as circles of approximately 15 μm wide in the pit membranes in bordered pit-pairs, as well as in the cross-field pitting of tracheids. The decrease in xyloglucan in the pit membranes corresponded to the occurrence of xyloglucanase activity in the stem which could be secreted by bacteria invading through cross-cut edges [16,17]. Such a phenomenon was also related to radiocesium movement, as shown by the autoradiographies in the series of sections. This is in agreement with our finding [6], which shows that by adding [137Cs]cesium chloride solution into poplar seedlings, [137]Cs flow could be markedly accelerated vertically through vessels, and horizontally through ray cells throughout the whole tissue if the xyloglucan content in the stem was reduced by using transgenic poplar overexpressing xyloglucanase. Although interaction might occur between radiocesium and acidic polymers in the membranes, immuno-cytochemical observation showed that the intervessel pit membranes of *Acer* are unlikely to contain pectic or other acidic polysaccharides [18].

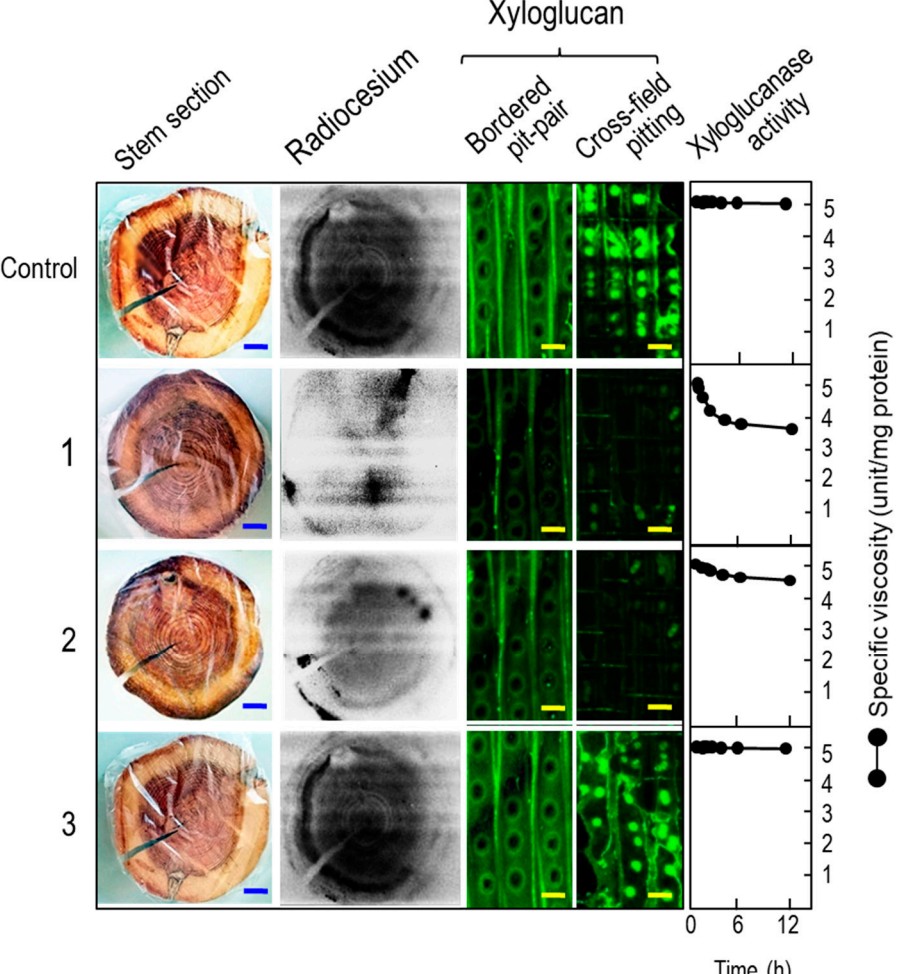

**Figure 3.** Ponding experiments for the stems of Japanese cedar from Fukushima forest. Analysis of the bottom edge of 165-day ponded stems that were drained by standing them vertically (dotted box in Figure 1). Bordered pit-pair and cross-field pitting were stained as xyloglucan with anti-xyloglucan antibody. Autoradiographies were taken from the insides of the cut surfaces. Xyloglucanase activity was determined by a decrease in the viscosity of tamarind xyloglucan. Scale: blue bar, 6 cm; yellow bar, 20 μm.

### 3.3. Lab-scale Ponding Experiments

To confirm xyloglucan as a target for the removal of radiocesium, small heartwood sections (16 cm$^2$ × 10 cm in length, 1960 Bq/kg) were sterilely removed from the stems and immersed in 25 °C water. After 84 days, see Figure 4, radiocesium had been decreased to 44% ± 26% of the level when immersed in Fukushima River water (37°33′27.0″ N 140°55′35.8″ E), compared with that of 97.2% ± 9.4% of the level in the case of immersion in distilled water. Nevertheless, a non-detectable level of radiocesium has been observed not only in the pool water used for 575 days but also in the water containing recombinant xyloglucanase (xyloglucan-specific endo-1,4-β-glucanase, twice with 1000 units). The decrease in xyloglucan was also confirmed by immuno-fluorescent staining of the tracheids. We conclude that the removal of radiocesium through water outflow can result from the degradation of xyloglucan in the walls of tracheids via the action of xyloglucanase-producing bacteria.

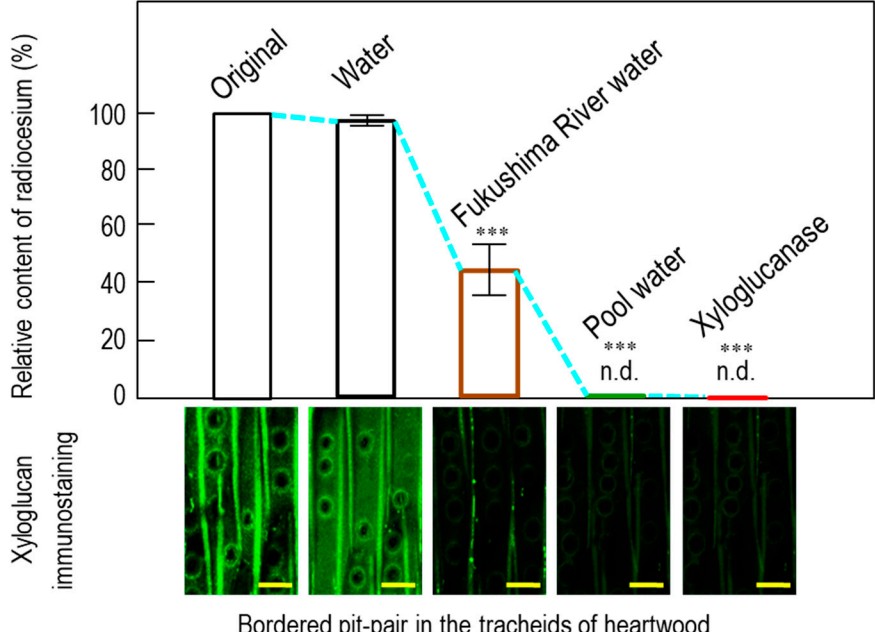

**Figure 4.** Effect of xyloglucanase on the removal of radiocesium during ponding in the heartwood section. Each tree sections (16 cm$^2$ × 10 cm in length) were ponded in each solution for 84 days. Immunostaining samples were obtained from the centers of each section. Significance at *** $p < 0.001$ levels compared with non-treated stems in Student's t-test are indicated. Scale bar, 20 μm.

### 3.4. Distribution of Radiocesium in Water Pool

After 575 days of ponding, the radiocesium level in the pool water rose to 0.2 Bq/kg, which could be approximately 30% of the radiocesium eluted from the stems, although the water in the pool contained non-detectable levels of radiocesium prior to the procedure. Since 20 kg of zeolite could have been absorbing radiocesium in limited amounts for 575 days [19], all the radiocesium eluted from the tree stems might not be recovered into the zeolite, as is evident in Table 1. Since radiocesium might also be absorbed into the concrete surfaces of the pool, it would not only be necessary to improve the adsorbing system of zeolite but also to ensure that the surface of the pool does not absorb radiocesium. In the case of a lab-scale ponding experiment for small heartwood sections, the radiocesium was almost completely recovered in 50 g of zeolite.

**Table 1.** Distribution of radiocesium in water and zeolite based on balance between elution and adsorption of radiocesium during ponding.

| Components | Radiocesium Level | | Level of Adsorption |
| --- | --- | --- | --- |
| | Day 0 | Day 575 | |
| | kBq | | % |
| Wood | 5100 | 2340 | – |
| Bark | 12,000 | 1 | – |
| Zeolite | 0 | 5400 | 31.5 |
| Water | 0 | 5200 | 30.4 |

It should be noted that this procedure could also remove radiocesium from the highly contaminated bark; levels of more than 300 kBq/kg dropped to 30 Bq/kg, as seen in Table 1, which could be useful for biomass energy applications.

## 4. Conclusions

Our study revealed that xyloglucan is the first target in the pit membranes, the degradation of which effectively aids in the removal of radiocesium from stem wood and accelerates water flow. Xyloglucan could keep the thinner microfibrils functioning as thin filters in the membranes because xyloglucan binds to cellulose through hydrogen and hydrophobic bonds [20–22]. The traditional technology of ponding can now be applied as a new biotechnology.

A better approach, therefore, would be to screen some bacteria producing strong xyloglucanase activity during ponding and employ them in a manner opposite to that used for copper uptake with *Bacillus licheniformis* in order to increase the permeability of spruce wood [23]. Nevertheless, it would also be easy to use local water containing bacteria.

**Supplementary Materials:** The following are available online at http://www.mdpi.com/1999-4907/11/5/589/s1, Figure S1: Attempted trials of potassium fertilization into forest trees by foliar spray either by hand (a) or radio-controlled helicopter (b and c) and by direct injection into tree trunks (d, e, and f) in a small forest. A 0.5 to 2 M KCl solution was used for the fertilization., Figure S2: Levels of radioactivity (cpm) determined by NaI (TI) scintillation counter (left) and those of radiation (μSv) using a Geiger counter (right), based on radiocesium content (Bq/kg) in the stem section., Figure S3: Ponding method for Fukushima forest trees. Cutting (a), immersion in the pool of water (b), stems in a pool (c), covered pool (d), vertical drain (e), and horizontal drain (f)., Table S1: Numbers of stems used for ponding method.

**Author Contributions:** T.S., R.A., S.M., S.Y., Y.K., and T.H. performed ponding experiments and A.M., R.A., S.M., S.Y., and T.H. performed the direct injection experiments in Fukushima. S.M., R.K., and K.B. performed the microscopy and T.S., T.T., Y.S., Y.K., T.I., and T.H. drafted the manuscript. All authors have read and agreed to the published version of the manuscript.

**Funding:** This work was financially supported through a grant from the Tokyo University of Agriculture for the Eastern Japan Reconstruction Support Project after the Fukushima Disaster and the MEXT-supported Program for the Strategic Research Foundation at Private Universities (S1311017).

**Acknowledgments:** We would like to thank S. Sasaki for allowing us to use his water pool with trees from his personal forests in Minamisoma. We are also thankful to E. Horiuchi and the Somachiho Forest Cooperative. We thank K. Omura for his technical support throughout the ponding experiments.

**Conflicts of Interest:** The authors declare no conflict of interest.

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
