# Peer review of "Intake of Radionuclides in the Trees of Fukushima Forests 3. Removal of Radiocesium from Stem Wood, Cryptomeria Japonica (L.f.) D. Don.†"

_forests, doi:10.3390/f11050589_

Round 1

Reviewer 1 Report

Forests 2020 II, 11

Article:  Intake of Radionuclides in the Trees …

  1. Seyama, R. Arakawa et al.

Comments for the authors

General remarks

In addition to the valuable information about flushing of radioactive cesium isotopes from the Japanese cedar wood by natural water, which is presumably of practical importance (biotechnology), the convincing indication that the degradation of xyloglucan in the pit membrane leads to efflux of those isotopes is important for basic research. Therefore, it is recommended to publish this manuscript also in its brief form. Further studies could look for bacteria producing great amounts of xyloglucanase.  Nevertheless, the text still needs a row of minor corrections before publication.

Detailed comments

 Title and Abstract

Line 2 and line 18: The scientific name of the tree species should appear in the Title and in the Abstract: Cryptomeria japonica (L.f.) D. Don.

Introduction

Lines 57-58: Sense is not quite clear. Do the authors perhaps mean: “It should be noted that the pit membrane is the target in the primary cell wall which is mainly composed of xyloglucan and cellulose.”  ?

Materials and Methods

Line 63:  How thick were the trunks on the average? Translocation rate presumably depends on the thickness and the age of the trees (Are they from the same age class?).

Results and Discussion

Lines 122-128: These lines should be omitted because they belong to “Material and Methods”. It is already described in lines 60-66 with similar words. (It should not be repeated here again.) Please start with line 128: “In the vertical stems…”

Page 4, Figure 1: It is not necessary to repeat the method in the legend because this was already described in “Material and Methods.”

Page 6, line 160: All Acer species world-wide? (In addition, it is known that the specific structure of xyloglucan differs between plant families.)

Page 6, Figure 1: It is recommended to put the number 2 underneath this graph and then renumber all following figures because the morphological and anatomical (pictures of anatomy) aspect is somewhat different from the other parts of Figure 1. (It is another additional story.)

Page 6, lines 164-165: This fits better to “Material and Methods” in the opinion of this reviewer.

Page 7, line 178: This reviewer thinks that the name of the absorbent is zeolite and not zeorite!

Page 8, legend of the Table:

  1. Again “Zeorite
  2. Also the information in the legend belongs more to “Material and Methods”.

        Conclusions

   What happens to the contaminated water after putting the wood in water? This question could be answered or discussed here or under “Results and Discussion”.

References were only checked by taking probes at random.

Line 243: Please check the name of the journal. It could be “Holz. Roh Werkstoff”.

Author Response

Dear Reviewer 1,

Thank you for your review.

Reviewer 2 Report

The manuscript “Intake of Radionuclides in the Trees of Fukushima Forests 3. Removal of radiocesium from stem wood“ addresses a method to flush out radiocesium from contaminated wood. According to the results from rather small number of replicates, the method is effective depending on the duration of the treatment and stem position. This is an interesting topic. However, the description of methods and the discussion part should be improved.

I have slight ethical concerns about the safety of involved researchers. Please add information on necessity and measures for protection against radiation.

Obviously, you combine results and discussion sections. However, your results should be discussed more intensely in perspective of previous studies and of your (still lacking) working hypotheses. Please refer to the journal’s guidelines as well. Consider to add a separate discussion section. Discuss limitations of your study as well e.g., small replication number.

Abstract: Too short, please add some details on results

Introduction:

- Please add detailed state of the art of methods to treat contaminated wood.

- Please add objectives, hypothesis and/ or research question.

L 63. Please rephrase “the trunks of which”
L 63. Some --> two?
L 67 – 69: Please rephrase, language.
L 69 – 70: I think this is not the right place for the sentence. Isn’t the water used in the paragraph above?

Methods section: Please state at the beginning of each subsection the purpose of the described procedure: “in order to….”. Please make the setup of the experiment clearer. For instance, it becomes clearer from figure caption 1A. Maybe you can use an additional figure (scheme) in methods section to clarify the setup.

Please discuss your small replication number / sample size. Is it similar in similar studies?

L122-128: This belongs to method section.
L135-136: Is this result or assumption? Consider moving the assumption to discussion section.

Section 3.2: Here you mix results and discussion, in my opinion.

L164 – 165: Useful for methods section.

Conclusions: Some parts of the conclusion could be shifted to discussion section.

Reference #1: Isn’t any peer-reviewed scientific publication available which you can use instead of Wikipedia?
Reference # 13 Klaassen et al. is not available. Please use a peer-reviewed reference.

Please change numbering of Fig. 1 A-C --> Fig 1, Fig 2, Fig 3

Fig 1B: Please use same scale of y-axis in all parts of the figure.

You put many demonstrative photos in the supplement. Consider to put some in the main part of the paper instead.

Author Response

Dear Reviewer 2,

Thank you for your review.

Round 2

Reviewer 2 Report

Dear Authors,

Your revision improved the paper distinctly. I have only some minor remarks. Thank you for explanation of the state of the art of contaminated wood treatment.

To ease the understand of your paper for other ignoramus like me, I would appreciate if you add just one sentence to introduction (maybe in line 55), saying exactly or similarly what you state in first sentence of your informative explanation: “There is no method to treat contaminated wood.”

Unfortunately, I do not find an answer to my remark concerning the safety of researchers. Please add a short statement to the methods section, stating e.g., that the exposure time is too short or the radiation level too low to be harmful or which protection measures were used.

Proposition for line 75: The heartwood sections (1960 Bq/kg) of the Japanese cedar stems were sterilely excised into small pieces (16 cm2 × 10 cm in length), mixed? / powdered? …? with 50 g of zeolite and….

Reference #13:
1. Apologies for confusion. When I click the links in first manuscript and in the revisied version these do not work. I receive only error messages. However, the link in your answer letter is working. Please check and provide the working link for the references of the final version of your paper.
2. In my opinion the paper is no peer-reviewed publication. If there is no more reliable publication on the issue it is OK to use it. Otherwise it would be better to cite a sound peer-reviewed publication from any recognized journal.

Typo in caption of Fig. 1, fourth line: **p<0.0,1  please remove the “,”
- Typo in caption of table 1 ZeoRite
- Typo in L 222: expeiment

L 69: Please rephrase. Proposition: The diameters of the stems were about 41 to 50 cm at breast height.

L 70-73: The explanation sounds slightly confusing to me. I think you can delete the sentence “After 165 and 575 days of ponding, some were removed from the pool and allowed to drain for 1 month.” Then, you can change the next sentence (proposition!): "Six of the 2-m stems (~1200 Bq/kg in the xylem) were removed from the pool after 165 days of ponding and six stems of same properties after 575 days. Stems of each subsample were allowed to drain for 1 month whereby three stems were positioned vertically while the other three were laid out horizontally."

Author Response

(The authors gave the same response as above.)
